# Impact of bone marrow involvement on outcome in relapsed and refractory transplant eligible diffuse large B-cell lymphoma and transformed indolent lymphoma

**Denis Terziev[1], Marcus Bauer[2], Lisa Paschold[1], Claudia Wickenhauser[2], Andreas Wienke[3], Mascha Binder[1], Lutz P. Müller[1], Thomas Weber[1]\***

1 Department of Internal Medicine IV, Haematology and Oncology, University Hospital Halle (Saale), Martin-Luther-University Halle-Wittenberg, Halle, Germany, 2 Institute of Pathology, University Hospital Halle (Saale), Martin-Luther-University Halle-Wittenberg, Halle, Germany, 3 Institute of Medical Epidemiology, Biometrics and Informatics, Martin-Luther-University Halle-Wittenberg, Halle (Saale), Halle, Germany

\* thomas.weber@uk-halle.de

**Data Availability Statement:** All relevant data are within the paper and its Supporting Information files.

## Abstract

In front-line treatment of diffuse large B-cell lymphoma (DLBCL), prior studies suggest that concordant but not discordant involvement of the bone marrow (BM) portends a poor prognosis. The prognostic impact of bone marrow infiltration (BMI) in recurrent or refractory DLBCL (r/rDLBCL) and transformed indolent lymphoma (r/rTRIL) patients is less clear. Thus, we examined the prognostic significance of the infiltration of bone marrow (BMI) by concordant, large B-cells (conBMI) and discordant, small B-cells (disBMI) in this patient group. We performed a single center retrospective analysis of the prognostic impact of BMI diagnosed before start of second-line treatment as well as multiple clinicopathologic variables in 82 patients with r/rDLBCL or r/rTRIL intended to treat with autologous SCT. Twenty-five of 82 patients (30.5%) had BMI. Out of these, 19 (76%) had conBMI and 6 (24%) had disBMI. In patients with conBMI but not disBMI, uni- and multivariate analysis revealed inferior progression free survival (PFS) and overall survival (OS) compared to patients without BMI (median PFS, 9.2 vs 17.45 months, log rank: p = 0.049; Hazard Ratio, 2.34 (Confidence Interval, 1.24–4.44), p = 0.009; median OS 14.72 vs 28.91 months, log rank: p = 0.017; Hazard Ratio, 2.76 (Confidence Interval, 1.43–5.31), p = 0.002). ConBMI was strongly associated with nonGCB subtype as classified by the Hans algorithm (82.4% vs 17.6%, p = 0.01). ConBMI comprised an independent predictor of poor prognosis in primary and secondary r/rDLBCL. Incorporating conBMI in the pretherapeutic risk assessment for r/rDLBCL and r/rTRIL patients may be useful for prognostication, for stratification in clinical trials, and to assess new therapies for this high-risk patient subset that might not benefit from SCT in second-line treatment.

**Funding:** The authors received no specific funding for this work.

**Competing interests:** The authors have declared that no competing interests exist.

## Introduction

Diffuse large B-cell lymphoma (DLBCL) is the most frequent type of lymphoma and is highly heterogeneous with regard to clinical manifestation, biological and molecular features and prognosis [1–3]. In eligible patients with refractory or recurrent DLBCL (r/rDLBCL) and transformed indolent lymphoma (r/rTRIL) the introduction of high-dose chemotherapy and autologous stem cell transplantation (SCT) following salvage immunochemotherapy led to long term survival rates of >50% [4,5]. Yet, up to 50% of initially transplant eligible patients are not able to receive autologous SCT due to failure of salvage therapy, failure of apheresis of autologous peripheral blood stem cells or therapy-limiting toxicity [5]. On the other hand, 40–50% of patients relapse within 4 years after autologous SCT [5–7], in that case resulting in very poor prognosis [8,9]. In primary refractory DLBCL and DLBCL relapsing within 12 months after completion of first line therapy, lymphoma-directed myeloablative conditioning followed by allogeneic SCT may result in a better outcome compared to autologous SCT. Still, OS 1 year post allogeneic SCT does not exceed 50% [10]. Thus, the benefit of autologous and allogeneic SCT in r/rDLBCL and r/rTRIL is ultimately limited and therefore, pretherapeutic risk stratification to identify a subset of patients who will not benefit from SCT and should therefore be referred to alternative treatment approaches is an urgent need in clinical practice.

5–22% of transplant eligible r/rDLBCL patients show lymphoma infiltration of the bone marrow (BM) [5,6,11,12]. In TRIL patients, up to 40% of BM involvement (BMI) has been reported [13]. While the BM is mostly concordantly involved by large cell lymphoma (con-BMI), a subset of positive BMI (posBMI) patients show BMI by discordant small cell lymphoma (disBMI). The latter represents a heterogeneous group of disorders, comprising cases with clonally related, as well as cases with two clonally distinct, unrelated B-cell neoplasms presenting synchronously as previously shown by comparative molecular analysis of immunoglobulin heavy chain (IgH) and BCL2 gene rearrangements [14]. Since BMI is clinically recognized as advanced disease, it contributes to higher International Prognostic Index (IPI) scores [15]. However, it has been reported that conBMI but not disBMI negatively impacts progression free survival (PFS) and overall survival (OS) in DLBCL patients after treatment with front-line therapy independently from the IPI [16–18]. Yet, owing to the dissimilarity of designs and inclusion criteria and the limited number of enrolled patients with BMI in studies concerning r/rDLBCL and r/rTRIL, the prognostic impact of conBMI and disBMI in this patient group is unknown. In addition, clinicopathologic characteristics of posBMI primary and secondary r/rDLBCL patients are so far not well investigated. We retrospectively investigated a patient cohort intended to treat with autologous SCT to address these questions and elucidate the role of BMI in transplant eligible r/rDLBCL and r/rTRIL patients.

## Patients and methods

This is a retrospective single center analysis of unselected patients with transplant eligible r/rDLBCL and r/rTRIL patients who were treated between November 1998 and March 2019 at the University Hospital Halle (Saale). Patients were identified by a review of the internal hospital database records and verified by using the transplant-team records and German Registry for Stem Cell Transplantation (DRST) database. TRIL was defined as the presence of indolent B-cell lymphoma prior to or at the time of DLBCL diagnosis [6]. Only patients with histologically proven transformation were included. Patients were included in the analysis if they were ≥18 years of age with a biopsy proven diagnosis of DLBCL according to the 2008 WHO classification [19], were intended to treat with a consolidating SCT and had complete clinical data. Exclusion criteria comprised primary involvement of the central nervous system, primary mediastinal B-cell lymphoma or coincident medical conditions that precluded treating the

lymphoma with curative intention. Assessed clinical data were age at relapse, sex, secondary age adjusted IPI (saaIPI) [20] (incorporating Ann Arbor staging >II, elevated LDH and ECOG performance score >1), TRIL, early relapse defined as duration of first complete remission <12 months or incomplete response following front-line treatment, bulky mass ≥7,5 cm in largest diameter at any time before start of induction therapy [21], secondary involvement of the central nervous system (CNS), prior Rituximab application, executed SCT, remission status before SCT and histologically confirmed BMI any time prior to start of induction therapy.

BM biopsy were centrally reviewed at the Institute of Pathology, Halle (Saale) by the pathologists MB and CW. ConBMI was defined as BMI with DLBCL, while disBMI was reported if the histological und immunohistochemical picture was compatible with small cell B-cell lymphoma. Flow cytometry, immunohistochemistry and/ or next generation sequencing (NGS) based analysis of IgH gene rearrangements were utilized to confirm the clonality of the B-cell population or an aberrant immunophenotype when BM involvement was clinically highly suspicious and/ or histology was inconclusive. All available lymphoma samples were immunohistochemically analyzed for GCB/ nonGCB status utilizing the classifier developed by Hans and colleagues [22]. In addition, available lymphoma tissue was immunohistochemically accessed for p53 alteration by IHC utilizing the p53 mouse monoclonal antibody (clone DO7, Cell Marque). A p53 overexpression and the lack of p53 immunohistochemical expression were considered positive [23]. Response to salvage therapy and, if applied, SCT was assessed by reviewing reports of contrasted computer tomography scans, magnetic resonance imaging (MRI) and/ or positron emission tomography with the tracer fluorine-18 fluorodeoxyglucose at time of relapse, after each line of salvage therapy and within three months after completion of SCT. Further, if BM was involved at any time before start of salvage therapy, remission assessment included the review of restaging BM biopsies. Response to salvage therapy was defined by achievement of at least partial remission (PR) defined by the diagnostic criteria of Lugano [24].

This study was performed according the Declaration of Helsinki and approved by the Ethics Committee at the Martin-Luther-University of Halle-Wittenberg (#2019–072).

## Statistical analyses

Clinical characteristics were compared between the separate BMI cohorts using the independent samples t-test for continuous variables and the $\chi^2$ test for categorical variables. Progression free survival (PFS) was defined as the time from disease recurrence until lymphoma progression or death of any cause. In a landmark analysis for patients who received SCT, PFS post SCT was defined as the time from SCT until lymphoma progression or death of any cause. Overall survival (OS) was defined as the time from disease recurrence until death from any cause and OS post SCT was calculated from SCT until death from any cause. Patients were censored at time of their last clinical visit by a physician. OS and PFS were assessed using the Kaplan-Meier (KM) method [25]. The log-rank test [26] was utilized for comparison of PFS and OS between groups. Multivariate analysis was performed using a Cox proportional hazards model [27] to assess the independent effect of prognostic variables on PFS and OS. A P value of ≤0.05 (two-sided) was considered statistically significant. We used SPSS version 25.0 for Windows; SPSS Inc, Chicago, IL for data analysis.

## Results

### Patient and treatment characteristics

A total of 82 patients with r/rDLBCL and r/rTRIL that met the inclusion criteria were identified. The median age of the entire cohort was 59 years (range, 23 to 75). At the time of analysis,

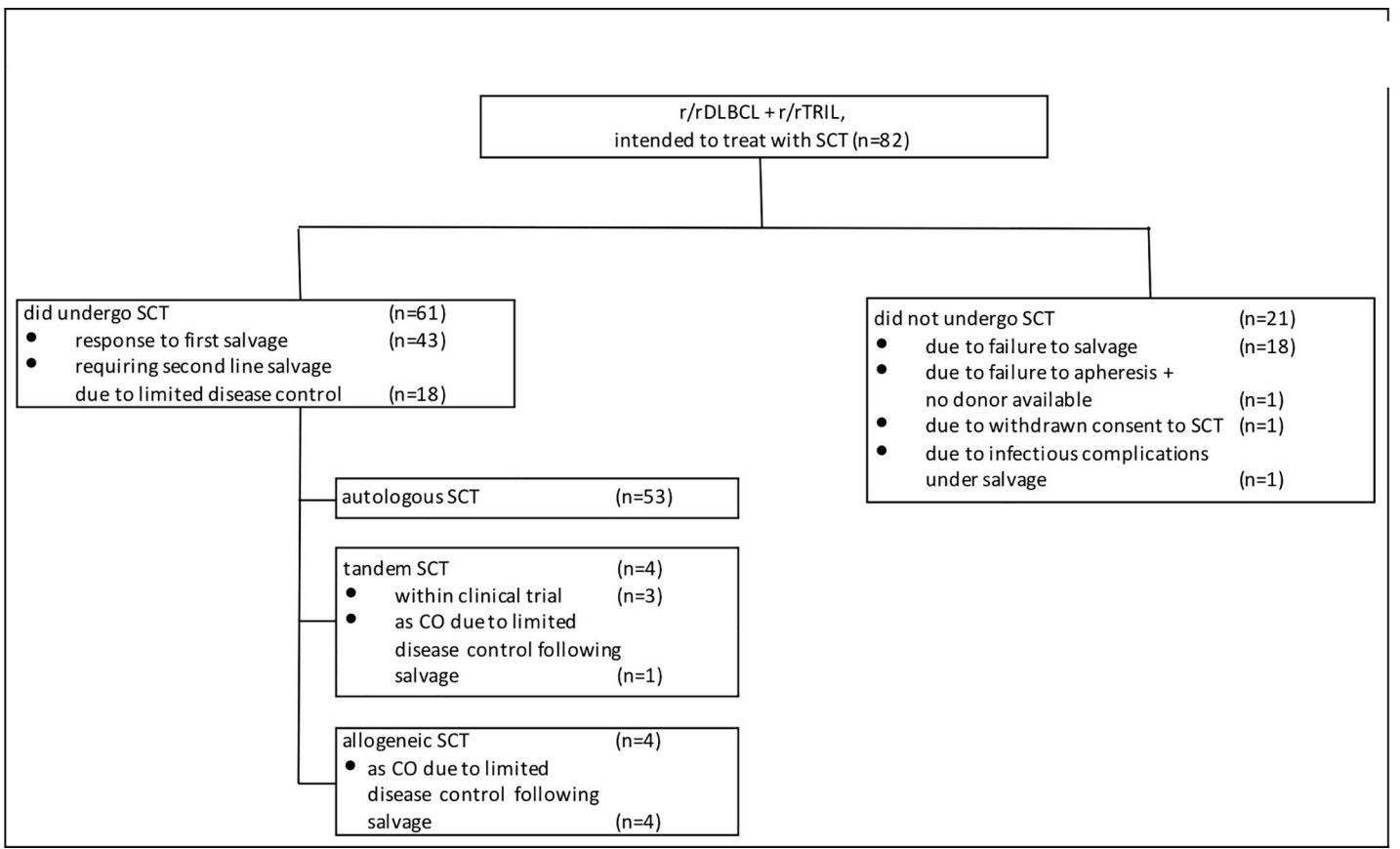

**Fig 1. Consort diagram for the present DLBCL and TRIL patient cohort intended to treat with SCT.** Abbreviations: r/r: recurrent or refractory, DLBCL: diffuse large B-cell lymphoma; SCT: stem cell transplantation, TRIL: transformed indolent lymphoma, CO: clinical option.

median follow-up time for living patients was 66.3 months (range, 8.1 to 238.7). Overall, 58 patients (70.7%) have died. All patients have received front-line anthracycline-containing chemotherapy (S1 Table). Front-line therapy contained Rituximab in 59 patients (72%). Salvage immunochemotherapy comprised (R)-DHAP [5] in 40 patients (48.8%), (R)-ICE [5] in 14 patients (17.1%), (R)-DexaBEAM [28] in 15 patients (18.3%) and others in 13 patients (15.9%). Autologous SCT was accomplished in 57 patients (69.5%) (Fig 1). Out of these, 4 patients (4.9%) had tandem autologous and allogeneic SCT. Highdose chemotherapy regimens foregoing autologous SCT comprised BEAM [29] in 35 patients (61.4%), TEC [30] in 11 patients (19.3%) and other regimens in 11 patients (19.3%). Four patients (4.8%), in which autologous SCT was not executed due to insufficient disease control (n = 3, 3.6%) or failure of stem cell apheresis after induction immunochemotherapy (n = 1, 1.2%), were alternatively referred to allogeneic SCT. Patient and treatment characteristics are given in Table 1.

## Characteristics of patients with BMI

A total of 25 patients (31.5%) had BMI at any time before start of induction therapy. Of these, 19 patients (23.2%) had conBMI and 6 patients (7.3%) had disBMI. Two patients with inconclusive BM histology were clinically suspicious for conBMI and showed a clonal B-cell population in BM accessed by NGS and were therefore assigned to the conBMI patient group. 57 patients (68.5%) had noBMI (noBMI). Table 1 summarizes the clinicopathological and therapy

**Table 1.  Clinicopathologic characteristics of patients grouped by type of BMI.**

| Chararacteristics | noBMI | conBMI | disBMI | P-value | | |
|---|---|---|---|---|---|---|
| | (n = 57) | (n = 19) | (n = 6) | | | |
| | No. (%) | No. (%) | No. (%) | | | |
| | | | | noBMI vs conBMI | noBMI vs disBMI | conBMI vs disBMI |
| Age at relapse (years) | | | | | | |
| Median (Range) | 59 (23–75) | 56 (36–71) | 58 (51–65) | 0.81 | 0.83 | 0.69 |
| Sex | | | | 0.381 | **0.041** | 0.199 |
| Male | 42 (73.7) | 12 (63.2) | 2 (33.3) | | | |
| Female | 15 (26.3) | 7 (36.8) | 4 (66.7) | | | |
| saaIPI ≥2 | 37 (64.9) | 16 (84.2) | 5 (83.3) | 0.113 | 0.363 | 0.959 |
| Bulky mass ≥7.5 cm | 21 (36.8) | 4 (21.1) | 2 (33.3) | 0.205 | 0.865 | 0.539 |
| Duration of first complete remission <12 months* | 41 (71.9) | 11 (57.9) | 2 (33) | 0.254 | 0.053 | 0.294 |
| TRIL | 11 (19.3) | 5 (26.3) | 6 (100) | 0.516 | **<0.001** | **0.002** |
| secondary CNS involvement | 12 (17.5) | 1 (5.3) | 0 (0) | 0.113 | 0.212 | 0.566 |
| prior Rituximab treatment | 44 (77.2) | 11 (57.9) | 4 (66.7) | 0.103 | 0.565 | 0.702 |
| failure to 1st salvage regimen | 22 (38.6) | 5 (26.3) | 3 (50) | 0.333 | 0.587 | 0.278 |
| completed SCT | 43 (75.4) | 14 (73.7) | 4 (66.7) | 0.878 | 0.639 | 0.739 |
| autologous SCT | 39 (68.4) | 11 (57.9) | 3 (50) | | | |
| allogeneic SCT | 2 (3.5) | 0 (0) | 2 (33.3) | | | |
| tandem SCT | 2 (3.5) | 1 (15.3) | 1 (16.7) | | | |
| no CR before SCT | 27 (62.8) | 8 (57.1) | 4 (100)** | 0.706 | 0.133 | 0.109 |
| no CR after SCT | 17 (40.5) | 6 (42.9) | 2 (50) | 0.875 | 0.712 | 0.8 |
| COO (IHC) | | | | **0.01** | 0.06 | 0.43 |
| GCB | 17 (56.7) | 3 (17.6) | 0 (0) | | | |
| nonGCB | 13 (43.3) | 14 (82.4) | 3 (100) | | | |
| unknown | 27 | 2 | 3 | | | |
| p53 expression (IHC) | | | | 0.088 | 0.184 | 0.859 |
| p53 wt | 15 (71.4) | 8 (44.4) | 2 (40) | | | |
| p53 alteration | 6 (28.6) | 10 (55.6) | 3 (60) | | | |
| unknown | 36 | 1 | 1 | | | |

Abbreviations: BMI: bone marrow infiltration, noBMI: no bone marrow infiltration, conBMI: concordant bone marrow infiltration, disBMI: discordant bone marrow infiltration, saaIPI: secondary age adjusted International Prognostic Index, TRIL: transformed indolent lymphoma, CNS: central nervous system, SCT: stem cell transplantation, CR: complete remission, COO: cell of origin, IHC: immunohistochemistry, GCB: germinal center B-cell, wt: wildtype

*including patients not achieving complete response after front-line treatment

**2 (50%) no response of indolent component, 2 (50%) no response of aggressive component.

associated characteristics for each of the three groups. Comparing all three groups, the noBMI, conBMI and disBMI group, we found no relevant difference in age, saaIPI, bulky disease, secondary CNS involvement, duration of first complete remission, prior Rituximab treatment, front-line regimens and completion of SCT. TRIL was more often diagnosed in patients with disBMI (disBMI 100% vs. conBMI 26.3% and noBMI 19.3%, p <0.01, respectively). The conBMI group was more likely to express a nonGCB immunophenotype as classified by the HANS algorithm compared to the noBMI group (82.4% vs. 43.3%, p = 0.01). Alteration of p53 was equally distributed between the BMI groups. The same observations were made when conBMI and disBMI were compared to the patient group with extensive disease (AA >2) and noBMI, respectively (S2 Table). An enrichment of secondary CNS involvement was only observed when comparing patients with AA >2 and noBMI to the conBMI group (AA>2, noBMI 27.3% vs. conBMI 5.3%, p = 0.048).

**Table 2. Prognostic factors of OS in r/rDLBCL and r/rTRIL patients, transplant eligible.**

| | total cohort (n = 82) | | | | | landmark analysis of the transplanted cohort (n = 61) | | | | |
|---|---|---|---|---|---|---|---|---|---|---|
| | Univariate analysis | | Multivariate analysis | | | Univariate analysis | | Multivariate analysis | | |
| Parameters | median OS (months) | P-value (log rank test) | HR | 95% CI | P-value | median PFS (months) | P-value (log rank test) | HR | 95% CI | P-value |
| Age at relapse | | | 1.01 | 0.99–1.04 | 0.319 | | | 1.01 | 0.98–1.03 | 0.635 |
| Sex | | 0.057 | 1.64 | 0.91–2.96 | 0.103 | | 0.661 | 1.19 | 0.56–2.53 | 0.643 |
| Male | 28.91 | | | | | 37.98 | | | | |
| Female | 11.89 | | | | | 23.79 | | | | |
| saaIPI | | **0.01** | 1.57 | 0.79–3.2 | 0.203 | | 0.1 | 1.53 | 0.7–3.32 | 0.286 |
| saaIPI <2 | 96.16 | | | | | 90.22 | | | | |
| saaIPI ≥2 | 12.78 | | | | | 14.1 | | | | |
| TRIL | | | 1.4 | 0.69–2.81 | 0.35 | | 0.185 | 1.72 | 0.73–4.04 | 0.213 |
| dnDLBCL | 30.23 | 0.089 | | | | 37.98 | | | | |
| TRIL | 12.78 | | | | | 14 | | | | |
| Bulky mass ≥7.5 cm | | **0.001** | 1.91 | 0.95–3.83 | 0.068 | | 0.072 | 1.53 | 0.61–3.83 | 0.362 |
| no | 36.7 | | | | | 37.98 | | | | |
| yes | 8.44 | | | | | 5.29 | | | | |
| Duration of first CR | | **0.021** | 2.27 | 1.07–4.81 | **0.032** | | 0.284 | 1.28 | 0.58–2.83 | 0.532 |
| <12 months* | 11.9 | | | | | 14 | | | | |
| ≥12 months | 43.96 | | | | | 38.87 | | | | |
| BMI | | **0.022** | | | | | **0.041** | | | |
| noBMI | 28.91 | | | | | 66.86 | | | | |
| posBMI | 13.54 | | | | | 14 | | | | |
| conBMI | 14.72 | **0.017** | 2.76 | 1.43–5.31 | **0.002** | 14 | **0.025** | 2.63 | 1.19–5.79 | **0.017** |
| disBMI | 11.89 | 0.4 | 1.47 | 0.42–5.14 | 0.543 | 7.1 | 0.587 | 0.62 | 0.12–3.23 | 0.572 |
| prior Rituximab treatment | | 0.315 | | | | | 0.627 | | | |
| no | 43.96 | | | | | 37.98 | | | | |
| yes | 13.54 | | | | | 28.39 | | | | |
| response to first salvage | | **0.006** | | | | | **<0.001** | | | |
| no | 9.3 | | | | | 4.34 | | | | |
| yes | 36.7 | | | | | 38.87 | | | | |
| undergone SCT | | **<0.001** | 0.313 | 0.16–0.63 | **0.001** | | | | | |
| no | 7 | | | | | | | | | |
| yes | 36.7 | | | | | | | | | |
| remission status before SCT | | | | | | | **0.008** | 2.64 | 1.19–5.87 | **0.017** |
| CR | | | | | | 82.46 | | | | |
| ≥PR | | | | | | 7.8 | | | | |
| COO (IHC) | | 0.379 | | | | | 0.78 | | | | |
| GCB subtype | 26.89 | | | | | 28.39 | | | | |
| nonGCB subtype | 18.86 | | | | | 25.79 | | | | |
| p53 expression (IHC) | | 0.451 | | | | | 0.191 | | | | |

*(Continued)*

**Table 2.** (Continued)

| | total cohort (n = 82) | | | | | landmark analysis of the transplanted cohort (n = 61) | | | | |
|---|---|---|---|---|---|---|---|---|---|---|
| | Univariate analysis | | Multivariate analysis | | | Univariate analysis | | Multivariate analysis | | |
| *Parameters* | *median OS (months)* | *P-value (log rank test)* | *HR* | *95% CI* | *P-value* | *median PFS (months)* | *P-value (log rank test)* | *HR* | *95% CI* | *P-value* |
| *p53 wt* | 19.09 | | | | | 28.39 | | | | |
| *p53 alteration* | 13.54 | | | | | 23.79 | | | | |

Abbreviations: OS: overall survival, r/r: recurrent or refractory, DLBCL: diffuse large B-cell lymphoma, TRIL: transformed indolent lymphoma, HR: Hazard Ratio, CI: Confidence Interval, saaIPI: secondary age adjusted International Prognostic Index, dnDLBCL: de novo diffuse large B-cell lymphoma, SCT: stem cell transplantation, BMI: bone marrow infiltration, noBMI: no bone marrow infiltration, posBMI: positive bone marrow infiltration, conBMI: concordant bone marrow infiltration, disBMI: discordant bone marrow infiltration, SCT: stem cell transplantation, CR: complete remission, PR: partial remission, COO: cell of origin, IHC: immunohistochemistry, GCB: germinal center B-cell, wt: wildtype

*including patients not achieving complete response after front-line treatment.

## Response to salvage therapy according to BMI

As expected, failure of salvage therapy in the entire cohort was significantly associated with early relapse (37% vs 7%, p = 0.004), bulky disease (52% vs 15%, p = 0.001) and female sex (46% vs 18%, p = 0.007). No relevant differences in response were found according to salvage chemotherapy regimens and if applied high-dose chemotherapy regimens foregoing autologous SCT. Also, patients who responded to salvage chemotherapy were more likely to proceed to SCT (92% vs 8%, p <0.001) (S3 Table).

Interestingly, no correlation of posBMI with response to salvage therapy was observed, as noBMI and posBMI patients showed a similar response rate (28% vs 26%, p = 0.874).

## Impact of BMI on survival

Table 2 and S4 Table summarize the results of all tested variables in uni- and multivariate survival analyses. BMI negatively affected survival (posBMI vs noBMI: median PFS 9.2 vs. 17.45 months, log-rank (KM) p = 0.055; median OS, 13.54 vs 28.91 months, log-rank (KM) p = 0.022; S1A Fig and S1B Fig). Importantly, conBMI portended a poor prognosis relative to noBMI (median PFS, 9.2 vs 17.45 months, log rank (KM) p = 0.049; median OS, 14.72 vs 28.91 months, log-rank (KM) p = 0.017, Fig 2A and 2B), whereas disBMI did not (median PFS, 6 vs 17.45 months, log rank (KM) p = 0.456; median OS, 11.89 vs 28.91; log rank (KM) p = 0.4; S1C Fig and S1D Fig). ConBMI maintained its prognostic relevance towards noBMI in the landmark PFS and OS analyses of the transplanted cohort (Fig 2C and 2D), in the extensive disease cohort (S1E Fig and S1F Fig) and in the nonGCB cohort (S1G Fig and S1H Fig).

In multivariate analysis utilizing Cox regression model of PFS and OS, conBMI and not performed SCT were adverse prognostic factors for both PFS and OS (PFS: S4 Table, OS: Table 2). Additionally, early relapse predicted poor OS in multivariate analysis. Sex, TRIL, saaIPI ≥2, bulky mass ≥7.5 cm and disBMI showed no prognostic relevant impact (Table 2). We also, separately, performed exploratory multivariate landmark analyses within the subgroup of transplanted patients, to determine if the risk factors for mortality or relapse were different in this group. In transplanted patients, non-achievement of complete remission (CR) before SCT was included into the multivariate analysis and was strongly associated with worse postSCT PFS and OS (S4 Table, Table 2). Of note, conBMI also maintained its prognostic relevance in multivariate landmark analysis in the transplanted patient subset (S4 Table, Table 2).

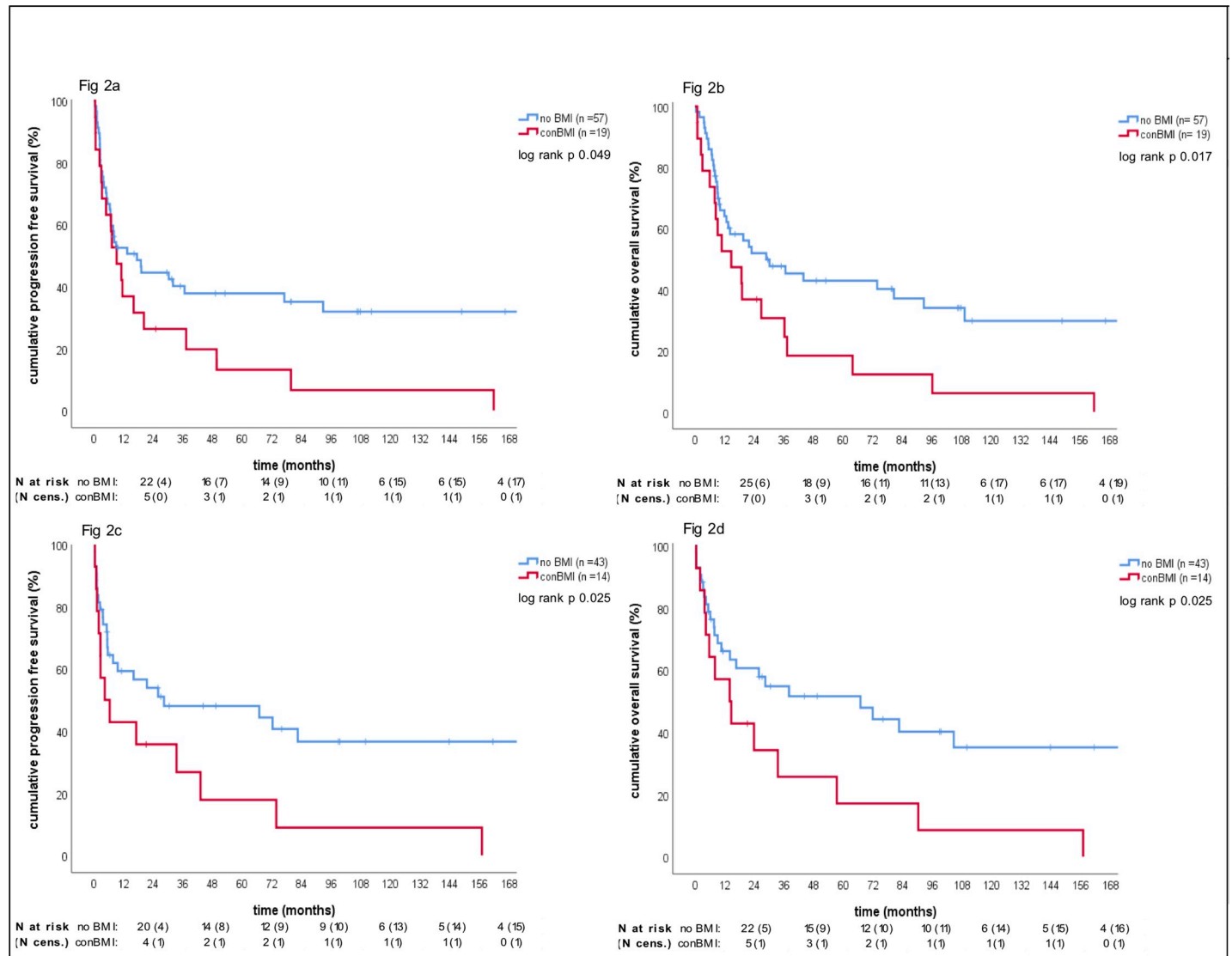

**Fig 2. Kaplan-Meier curves for progression free survival and overall survival according to different groups of BMI.** a: KM curve for PFS according to noBMI vs conBMI in the total cohort; b: KM curve for OS according to noBMI vs conBMI in the total cohort; c: KM curve for PFS landmark analysis according to noBMI vs conBMI in the transplanted patient subset; d: KM curve of OS landmark analysis according to noBMI vs conBMI in the transplanted patient subset; Abbreviations: BMI: bone marrow infiltration, noBMI: no bone marrow infiltration, conBMI: concordant bone marrow infiltration, KM: Kaplan-Meier; PFS: progression free survival, OS: overall survival, N: number of patients, cens.: censored.

## Discussion

To the best of our knowledge, we present the first detailed analysis of the prognostic impact of BMI on outcome of transplant eligible r/rDLBCL and r/rTRIL patients. In this analysis, conBMI but not disBMI was associated with worse PFS and OS in transplant eligible patients. This association was irrespective of Ann Arbor stage and saaIPI indicating conBMI might not only be a measure of the extend of the disease.

As reported previously [4–6,31,32], early relapse, achievement of CR preceding SCT and completion of SCT, which was strongly associated with response to the salvage therapy (91.2% vs 4%, p <0.001), displayed strong factors for PFS and OS in our analyses. Taken together, these parameters emphasize the main limitation of the SCT strategy since response to

chemotherapy remains a fundamental prerequisite for SCT. Further parameters, which were able to discriminate subgroups of patients with differing prognosis in our study include saaIPI ≥2 [20,33] and bulky disease [21,34]. However, while these parameters impacted survival in univariate analyses, they failed to show relevant influence on PFS and OS in multivariate analyses of the transplanted subgroup if achievement of CR before SCT was included into the model. This suggests that these factors indicate the likelihood of achieving CR after salvage immunochemotherapy (S3 Table), rather than affecting the outcome of SCT for patients who do reach CR [6]. In contrast to these findings, conBMI but not disBMI portended worse PFS and OS compared to the noBMI group independently from other prognostically relevant factors.

In analogy to findings in first diagnosed DLBCL [35], alteration of p53 expression, which is a known marker for high clinical stage [36] but not for adverse survival [37], was not associated with BMI subgroups in the present cohort. On the other hand, conBMI was significantly associated with nonGCB subtype as classified by the Hans-algorithm [22], which is in opposition to findings in front-line setting of DLBCL [35]. Yet, the prognostic impact of conBMI appears to be independent from the cell of origin subtype as it maintained its prognostic relevance both, in the GCB and the in the nonGCB cohort. Recently, Yao et al. [35] reported on broader biological characteristics in initially diagnosed DLBCL patients, revealing that conBMI also shows an enrichment for unfavorable markers such as CD5 expression [38] and MYC gene rearrangement [39]. Yet, of interest, conBMI remained an adverse predictor in almost all tested biomarker-positive DLBCL subsets. The authors thus assumed that conBMI may not be entirely a surrogate for these known adverse biological features in DLBCL, but other possible mechanisms underlying conBMI may contribute to adverse survival. In their study, gene expression profiling of DLBCL specimen in the reported cohort revealed that conBMI in comparison to noBMI additionally shows an upregulation of genes encoding various proteins playing a key role in cellular adhesion or cytoskeletal reorganization as well as immunoregulation [35]. Therefore, conBMI may be related to cellular adhesion or migration and immune tolerance or escape within the BM niche. Further, adhesion of DLBCL to stroma cells within the BM niche could explain drug resistance and therefore limited prognosis [40].

The results from our studied patient cohort, which is comparable in terms of basic patient and disease characteristics to transplant eligible r/rDLBCL patients treated within prior prospective trials [5,31,41] underline the hypothesis that conBMI expresses an easy to access surrogate for an underlying unfavorable lymphoma biology in the second-line setting of DLBCL and TRIL. However, the current results are subject to several limitations mainly caused by the retrospective character of this analysis and the heterogeneity of the described cohort. Further, the relatively small patient size increases the risk of statistical errors.

Nonetheless, based on the presented data, we consider conBMI to be a relevant and independent clinical marker for poor prognosis in r/rDLBCL and r/rTRIL patients. Since incorporating conBMI in the pretherapeutic assessment could improve our ability to risk stratify patients with r/rDLBCL considering SCT, we hope that the results presented here may be further validated and enable precise prognostication and individualized disease surveillance as well as contribute to the design of upcoming clinical trials. Moreover, as the mechanisms underlying conBMI are not fully elucidated, further investigation of the biological background of DLBCL with conBMI is needed to identify potential therapeutic targets.

## Supporting information

**S1 Fig. Kaplan-Meier curves for progression free survival and overall survival according to different groups of BMI.** a: KM curve for PFS according to noBMI vs BMI in the total cohort;

b: KM curve for OS according to noBMI vs BMI in the total cohort; c: KM curve for PFS according to noBMI vs disBMI in the total cohort; d: KM curve of OS acc ording to noBMI vs disBMI in the total cohort; e: KM curve for PFS according to noBMI vs conBMI in the extensive disease (AA>2) patient subset, f: KM curve for OS according to noBMI vs conBMI in the extensive disease (AA>2) patient subset; g: KM curve for PFS according to noBMI vs conBMI in the nonGCB patient subset; h: KM curve for OS according to noBMI vs conBMI in nonGCB patient subset; Abbreviations: BMI: bone marrow infiltration, noBMI: no bone marrow infiltration, conBMI: concordant bone marrow infiltration, KM: Kaplan-Meier; PFS: progression free survival, OS: overall survival, AA: Ann Arbor, GCB: germinal center B-cell.
(TIFF)

**S1 Table. Front-line regimens of patients grouped by type of BMI.** Abbreviations: BMI: bone marrow infiltration, noBMI: no bone marrow infiltration, conBMI: concordant bone marrow infiltration, disBMI: discordant bone marrow infiltration, CHOP: cyclophosphamide, hydroxydaunorubicin, vincristine, prednisolone, R: rituximab, CHOEP: cyclophosphamide, hydroxydaunorubicin, vincristine, etoposide, prednisolone, DA-EPOCH-R: dose adjusted etoposide, prednisolone, vincristine, cyclophosphamide, doxorubicin, rituximab.
(XLSX)

**S2 Table. Clinicopathologic characteristics of patients with extensive disease defined as AA >2 grouped by type of BMI.** Abbreviations: AA: Ann Arbor stage, BMI: bone marrow infiltration, noBMI: no bone marrow infiltration, conBMI: concordant bone marrow infiltration, disBMI: discordant bone marrow infiltration, saaIPI: secondary age adjusted International Prognostic Index, TRIL: transformed indolent lymphoma, CNS: central nervous system, SCT: stem cell transplantation, CR: complete remission, COO: cell of origin, IHC: immunohistochemistry GCB: germinal center B-cell, wt: wildtype; *including patients not achieving complete response after front-line treatment.
(XLSX)

**S3 Table. Best response to salvage therapy without SCT according to prognostic factors.** Abbreviations: SCT: stem cell transplantation, CR: complete remission, PR: partial remission, SD: stable disease, PD: progressive disease, BMI: bone marrow infiltration, noBMI: no bone marrow infiltration, posBMI: positive bone marrow infiltration, saaIPI: secondary age adjusted International Prognostic Index, TRIL: transformed indolent lymphoma, dnDLBCL: de novo diffuse large B-cell lymphoma, SCT: stem cell transplantation, CTx: chemotherapy, COO: cell of origin, IHC: immunohistochemistry, GCB: germinal center B-cell, wt: wildtype; *response to autologous SCT.
(XLSX)

**S4 Table. Prognostic factors of PFS in r/rDLBCL and r/rTRIL patients, transplant eligible.** Abbreviations: PFS: progression free survival, r/r: recurrent or refractory, DLBCL: diffuse large B-cell lymphoma, TRIL: transformed indolent lymphoma, HR: Hazard Ratio, CI: Confidence Interval, saaIPI: secondary age adjusted International Prognostic Index, dnDLBCL: de novo diffuse large B-cell lymphoma, SCT: stem cell transplantation, BMI: bone marrow infiltration, noBMI: no bone marrow infiltration, posBMI: positive bone marrow infiltration, conBMI: concordant bone marrow infiltration, disBMI: discordant bone marrow infiltration, SCT: stem cell transplantation, CR: complete remission, CR: complete remission, PR: partial remission, COO: cell of origin, IHC: immunohistochemistry, GCB: germinal center B-cell; *including patients not achieving complete response after front-line treatment.
(XLSX)

## Author Contributions

**Conceptualization:** Thomas Weber.

**Data curation:** Denis Terziev, Lutz P. Müller, Thomas Weber.

**Formal analysis:** Denis Terziev, Marcus Bauer, Lisa Paschold, Claudia Wickenhauser, Andreas Wienke, Lutz P. Müller, Thomas Weber.

**Investigation:** Denis Terziev, Lisa Paschold, Claudia Wickenhauser, Thomas Weber.

**Methodology:** Andreas Wienke, Mascha Binder, Thomas Weber.

**Supervision:** Thomas Weber.

**Validation:** Andreas Wienke, Thomas Weber.

**Visualization:** Denis Terziev.

**Writing – original draft:** Denis Terziev, Thomas Weber.

**Writing – review & editing:** Denis Terziev, Marcus Bauer, Lisa Paschold, Claudia Wickenhauser, Andreas Wienke, Mascha Binder, Lutz P. Müller, Thomas Weber.

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
