## [Decision Letter · Decision Letter 0]

8 Apr 2020

PONE-D-20-07437

Impact of Bone Marrow Involvement on Outcome in relapsed and refractory transplant eligible Diffuse Large B-Cell Lymphoma and Transformed Indolent Lymphoma

PLOS ONE

Dear Dr. Weber,

Thank you for submitting your manuscript to PLOS ONE. After careful consideration, we feel that it has merit but does not fully meet PLOS ONE’s publication criteria as it currently stands. Therefore, we invite you to submit a revised version of the manuscript that addresses the points raised during the review process by both reviewers, experts in the lymphoma field.

We would appreciate receiving your revised manuscript by May 23 2020 11:59PM. To enhance the reproducibility of your results, we recommend that if applicable you deposit your laboratory protocols in protocols.io, where a protocol can be assigned its own identifier (DOI) such that it can be cited independently in the future. For instructions see: http://journals.plos.org/plosone/s/submission-guidelines#loc-laboratory-protocols

We look forward to receiving your revised manuscript.

Kind regards,

Francesco Bertolini, MD, PhD

Academic Editor

PLOS ONE

Journal Requirements:

2. Please provide a sample size and power calculation in the Methods, or discuss the reasons for not performing one before study initiation.

Reviewers' comments:

Reviewer's Responses to Questions

**Comments to the Author**

1. Is the manuscript technically sound, and do the data support the conclusions?

Reviewer #1: Yes

Reviewer #2: Yes

2. Has the statistical analysis been performed appropriately and rigorously? 

Reviewer #1: Yes

Reviewer #2: Yes

3. Have the authors made all data underlying the findings in their manuscript fully available?

Reviewer #1: Yes

Reviewer #2: Yes

4. Is the manuscript presented in an intelligible fashion and written in standard English?

Reviewer #1: Yes

Reviewer #2: Yes

5. Review Comments to the Author

Reviewer #1: This paper contributes to the field of prognostic factors in relapsed or refractory (R/R) diffuse large B cell (DLBCL) or transformed indolent (TRIL) non Hodgkin lymphoma. This is a retrospective observational single center study on the prognostic role of bone marrow involvement (BMI) by large B cell (concordant involvement) or by small cell (discordant involvement) lymphoma in 82 patients with transplant eligible recurrent or refractory DLBCL or TRIL treated with salvage chemotherapy over 20 years (from 1998 to 2019). The study clarifies that concordant but not discordant bone marrow involvement was responsible for a lower progression-free and overall survival in the whole patient population and in the subset of those patients who were able to receive a transplant. The authors recognize the limits of their study, mainly related to the retrospective analysis and to the small patient population of each subgroup (no BMI, n=57, vs concordant BMI, n=19, vs discordant BMI, n=6). Nevertheless, the observation that patients presenting with concordant BMI had 2.34 risk to fail salvage treatment, with or without autologous stem cell or allogeneic transplant, and 2.77 risk to die compared to their counterpart without BMI or with discordant BMI may help change clinical practice in this poor prognosis subgroup of R/R DLBCL or TRIL and therefore the study is worthwhile to be published.

Major Comments:

Abstract, page 2: Authors should include a definition of concordant and discordant BMI also in the abstract section

Patients and Methods pag 4: Authors should report how patients were selected: from a database, from a review of medical records, a review of all the histological diagnosis of NHL; how many patients with DLBCL were treated with frontline and/or salvage therapy in the same period?

Results, Patient and treatment characteristics, pag 6: Authors should report in details the front-line regimens used in a supplemental table

Results, response to salvage therapy, page 10: In S2 table patients with refractory disease and those with relapse within 12 months should be listed and analyzed separately.

Was there any difference in response rate based on salvage induction chemotherapy or myeloablative regimen?

Figure 2 a-d: Authors should report number of patients at risk

Reviewer #2: - In Table 1 remission duration < 12 month specify first remission

- Specify if in Table 1 no CR before SCT for dis BMI was due to non response of aggressive component, indolent component or both

- Some more information could be interesting:

1. any data about p53 disruption in all three groups? Expecially for disBMI

2. it could be important to have data about IGH rearragment in trasformed lymphoma, same rearragment and two different clone may have different outcome.

3. any data about FISH analysis for MYC, BCL2 and BCL6.

4. any data about risk of CNS involvement between no BMI an BMI?

6. PLOS authors have the option to publish the peer review history of their article (what does this mean?). If published, this will include your full peer review and any attached files.

Reviewer #1: Yes: Simonetta Viviani

Reviewer #2: No

---

## [Author Response · Author response to Decision Letter 0]

2 Jun 2020

In general:

We identified two minor mistakes in our statistics. We corrected all affected parameters in the revised manuscript. No relevant changes in significance levels occurred.

We corrected some spelling errors. 

Reviewer 1:

1. Abstract, page 2: Authors should include a definition of concordant and discordant BMI also in the abstract section:

We included the definition in the abstract on page 2 lines 6-9.

2. Patients and Methods page 4: Authors should report how patients were selected: from a database, from a review of medical records, a review of all the histological diagnosis of NHL:

Patients were identified from a review of medical records in the internal hospital database. Patients were verified by using the transplant-team records and German Registry for Stem Cell Transplantation (DRST) database. We identified 82 patients and included all in the analyses.

We added a statement in the methods section on page 4 lines 19-21.

3. How many patients with DLBCL were treated with frontline and/or salvage therapy in the same period?

We thank the reviewer for this important question. To the best of our knowledge, we included all transplant eligible patients of this time interval in our analysis.

We double-checked the number of patients by tracking the ICD10 codes for DLBCL retrospectively in our electronic inhouse database from 2015 to 2019 (when ICD-10 tracking was available). Using this method, we identified 126 patients with DLBCL or TRIL from 2014 to 2019 who received treatment at our institution. Out of these 73 patients (59%) received solely front-line treatment and 52 patients with rrDLBCL and rrTRIL (41%) received salvage immunochemotherapy. 32 of 52 patients with rrDLBCL or rrTRIL (62%) were considered initially transplant eligible. All 32 were included in our analysis. Thus, we assume that a representative proportion of patients with rrDLBCL and rrTRIL was included overall in our study.

4. Results, Patient and treatment characteristics, pag 6: Authors should report in details the front-line regimens used in a supplemental table.

We added the front-line regimens in the new table S1.

5. Results, response to salvage therapy, page 10: In S2 table patients with refractory disease and those with relapse within 12 months should be listed and analyzed separately.

We thank the reviewer for pointing to this important aspect. We added this information now in table S3.

6. Was there any difference in response rate based on salvage induction chemotherapy or myeloablative regimen?

No relevant differences in response were found according to salvage chemotherapy regimens and if applied high-dose chemotherapy regimens foregoing autologous SCT. We added information on response rates based on salvage induction and myeloablative regimen foregoing autologous SCT in table S3 and within the manuscript in Results, page 11 line 25 to page12 line 3. 

7. Figure 2 a-d: Authors should report number of patients at risk.

We added the number of patients at risk in figure 2a-d.

Reviewer 2:

1. In Table 1 remission duration < 12 months specify first remission.

We thank the reviewer for pointing out this important aspect. In the current work, we defined remission according to the CORAL stud by Gisselbrecht et al. (J Clin Oncol. 2010;28:4184–90): a) in complete remission after front line treatment, relapsed after 12 months, b) in complete remission after front line treatment, relapsed within 12 months, c) no achievement of complete remission after front line treatment (incomplete response). In our analysis we compared patient group a) vs. b) and c) (“early relapse”), since they show similar impact on survival after completed autologous SCT as shown within the PARMA trial (Philip et al.; N Engl J Med. 1995;333:1540–5.) We now specified the remission duration in table 1 (duration of first complete remission <12 months) and its legend (“*including patients not achieving complete response after front-line treatment”. We also specified this aspect in the Methods section page 5, line 6-8 (“early relapse defined as duration of first complete remission <12 months after completed front-line treatment including incomplete remission).

2. Specify if in Table 1 no CR before SCT for dis BMI was due to non-response of aggressive component, indolent component or both:

We specified the reason of not achieving complete remission in the legend of table 1: “2 (50%) no response of indolent component, 2 (50%) no response of aggressive component”.

3. Some more information could be interesting:

3.1. Any data about p53 disruption in all three groups? Expecially for disBMI.

As Sanger sequencing was unavailable due to institutional COVID19-regulations, we performed p53-expression analyses by routine automatized immunohistochemical assessment using the Cell Marque anti-p53-antibody clone DO7. A reduced expression or overexpression was defined as altered TP53-status. We added this in Methods on page 5, line 22-25. In our present cohort, alteration of p53 expression was not associated with BMI subgroups and did not affect survival. We added this information within table 1-2, S2-4 table and the discussion section page 17, line 23 to page 18 line 1.

3.2. It could be important to have data about IGH rearrangement in transformed lymphoma, same rearrangement and two different clone may have different outcome.

We thank the reviewer for bringing up this very interesting aspect. Unfortunately, we were not able to perform NGS on TRIL tissue during the given time due to institutional COVID-19 regulation. We hope to perform this analysis in nearby future.

3.3. Any data about FISH analysis for MYC, BCL2 and BCL6.

This is an important point. We performed MYC, BCL2 and BCL6 breakpoint analyses by CISH. Unfortunately, we did not obtain utilizable results. Inter alia, this is due to the FFPE-material with unbuffered paraffin used in our institution until 2010.

3.4. Any data about risk of CNS involvement between no BMI and BMI?

We thank the reviewer for making this point. An enrichment of secondary CNS involvement was only observed when comparing patients with extensive disease and noBMI vs. conBMI (AA>2, noBMI (n=12, 27.3%) vs. conBMI (n=1, 5.3%, p =0.048). This is in opposition to the association of BMI and risk of CNS relapse in the first line setting. However, in opinion numbers are too small to draw further conclusions. We included secondary CNS involvement in Methods, page 5, lines 9-10. The distribution of CNS involvement between patient groups was included in table 1 and in Results page 11 lines 18-20.

---

## [Decision Letter · Decision Letter 1]

23 Jun 2020

Impact of Bone Marrow Involvement on Outcome in relapsed and refractory transplant eligible Diffuse Large B-Cell Lymphoma and Transformed Indolent Lymphoma

PONE-D-20-07437R1

Dear Dr. Weber,

We’re pleased to inform you that your manuscript has been judged scientifically suitable for publication and will be formally accepted for publication once it meets all outstanding technical requirements.

Kind regards,

Francesco Bertolini, MD, PhD

Academic Editor

PLOS ONE

Additional Editor Comments (optional):

Reviewers' comments:

Reviewer's Responses to Questions

**Comments to the Author**

1. If the authors have adequately addressed your comments raised in a previous round of review and you feel that this manuscript is now acceptable for publication, you may indicate that here to bypass the “Comments to the Author” section, enter your conflict of interest statement in the “Confidential to Editor” section, and submit your "Accept" recommendation.

Reviewer #1: All comments have been addressed

Reviewer #2: All comments have been addressed

2. Is the manuscript technically sound, and do the data support the conclusions?

Reviewer #1: (No Response)

Reviewer #2: Yes

3. Has the statistical analysis been performed appropriately and rigorously? 

Reviewer #1: (No Response)

Reviewer #2: Yes

4. Have the authors made all data underlying the findings in their manuscript fully available?

Reviewer #1: (No Response)

Reviewer #2: Yes

5. Is the manuscript presented in an intelligible fashion and written in standard English?

Reviewer #1: (No Response)

Reviewer #2: Yes

6. Review Comments to the Author

Reviewer #1: (No Response)

Reviewer #2: Thank you for updating the paper. I found the additional informations quites interesting. I've no other questions

7. PLOS authors have the option to publish the peer review history of their article (what does this mean?). If published, this will include your full peer review and any attached files.

Reviewer #1: Yes: Simonetta Viviani

Reviewer #2: No

---

## [Editor Report · Acceptance letter]

26 Jun 2020

PONE-D-20-07437R1 

Impact of Bone Marrow Involvement on Outcome in relapsed and refractory transplant eligible Diffuse Large B-Cell Lymphoma and Transformed Indolent Lymphoma 

Dear Dr. Weber:

I'm pleased to inform you that your manuscript has been deemed suitable for publication in PLOS ONE. Congratulations! Your manuscript is now with our production department. 

Kind regards, 

on behalf of

Dr. Francesco Bertolini 

Academic Editor

PLOS ONE